# Combination of Resminostat with Ruxolitinib Exerts Antitumor Effects in the Chick Embryo Chorioallantoic Membrane Model for Cutaneous T Cell Lymphoma

**DOI:** 10.3390/cancers14041070

**Published:** 2022-02-20

**Authors:** Fani Karagianni, Christina Piperi, Berta Casar, Dalia de la Fuente-Vivas, Rocío García-Gómez, Kyriaki Lampadaki, Vasiliki Pappa, Evangelia Papadavid

**Affiliations:** 1National Center of Rare Diseases-Cutaneous Lymphoma—Member of EuroBloodNet, Second Department of Dermatology and Venereal Diseases, Attikon University General Hospital, National and Kapodistrian University of Athens, 124 62 Athens, Greece; karagiannifani@gmail.com (F.K.); sundylam@yahoo.gr (K.L.); 2Department of Biological Chemistry, Medical School of Athens, National and Kapodistrian University of Athens, 115 27 Athens, Greece; cpiperi@med.uoa.gr; 3Instituto de Biomedicina y Biotecnología de Cantabria, Consejo Superior de Investigaciones Científicas (CSIC)-Universidad de Cantabria, 39011 Santander, Spain; daliadelafuentevivas@gmail.com (D.d.l.F.-V.); rocio.garciagomez@unican.es (R.G.-G.); 4Centro de Investigación Biomédica en Red de Cáncer (CIBERONC), Instituto de Salud Carlos III, 28029 Madrid, Spain; 52nd Department of Internal Medicine—Propaedeutic and Research Unit, National and Kapodistrian University of Athens, Medical School of Athens, University General Hospital Attikon, 124 62 Athens, Greece; vas_pappa@yahoo.com

**Keywords:** chicken chorioallantoic membrane, CAM, MyLa, SeAx, Resminostat, Ruxolitinib, JAKi, HDACi, intravasation, extravasation, metastasis

## Abstract

**Simple Summary:**

The combination of Resminostat (HDACi) and Ruxolitinib (JAKi) exerted cytotoxic effects and inhibited proliferation of CTCL cell lines (MyLa, SeAx) in vitro. The aim of the present study was to validate their antitumor effects in vivo using the chick embryo chorioallantoic membrane (CAM) model, which allows quick and efficient monitoring of tumor growth, migration, invasion, and metastatic potential. The drug combination exhibited a significant inhibition of primary tumor size, and inhibited intravasation and extravasation of tumor cells to the liver and lung. It also exerted an inhibitory effect in the migration and invasion of tumor cells and significantly reduced key signaling pathway activation. Our data demonstrate that the CAM assay could be employed as a preclinical in vivo model in CTCL for pharmacological testing, and that the combination of Resminostat and Ruxolitinib exerts significant antitumor effects in CTCL progression that need to be further evaluated in a clinical setting.

**Abstract:**

The combination of Resminostat (HDACi) and Ruxolitinib (JAKi) exerted cytotoxic effects and inhibited proliferation of CTCL cell lines (MyLa, SeAx) in previously published work. A xenograft tumor formation was produced by implanting the MyLa or SeAx cells on top of the chick embryo chorioallantoic membrane (CAM). The CAM assay protocol was developed to monitor the metastatic properties of CTCL cells and the effects of Resminostat and/or Ruxolitinib in vivo. In the spontaneous CAM assays, Resminostat and Ruxolitinib treatment inhibited the cell proliferation (*p* < 0.001) of MyLa and SeAx, and induced cell apoptosis (*p* < 0.005, *p* < 0.001, respectively). Although monotherapies reduced the size of primary tumors in the metastasis CAM assay, the drug combination exhibited a significant inhibition of primary tumor size (*p* < 0.0001). Furthermore, the combined treatment inhibited the intravasation of MyLa (*p* < 0.005) and SeAx cells (*p* < 0.0001) in the organs, as well as their extravasation to the liver (*p* < 0.0001) and lung (*p* < 0.0001). The drug combination also exerted a stronger inhibitory effect in migration (*p* < 0.0001) rather in invasion (*p* < 0.005) of both MyLa and SeAx cells. It further reduced p-p38, p-ERK, p-AKT, and p-STAT in MyLa cells, while it decreased p-ERK and p-STAT in SeAx cells in CAM tumors. Our data demonstrated that the CAM assay could be employed as a preclinical in vivo model in CTCL for pharmacological testing. In agreement with previous in vitro data, the combination of Resminostat and Ruxolitinib was shown to exert antitumor effects in CTCL in vivo.

## 1. Introduction

Cutaneous T-cell Lymphomas (CTCLs) present a diverse group of extra-nodal non-Hodgkin lymphomas characterized by clonal growth of malignant T cells in the skin [1]. The most common type of CTCL is Mycosis Fungoides (MF), which is characterized by patches, infiltrated plaques, or tumors [2], whereas Sezary Syndrome (SS) is characterized by erythroderma, lymphadenopathy, and the presence of a malignant T cell clone in the peripheral blood and skin [3,4]. MF/SS are incurable and generally lethal in advanced stages [5], characterized by a chronic, relapsing course that necessitates repeat treatment regimens [6]. It is critical to establish new regimens for MF/SS patients with long-lasting and tolerable responses. Therapeutic strategies that combine novel agents with current treatment options may prove beneficial in the future management of CTCL patients [7]. To this end, there is a need for better understanding of the major biological mechanisms underlying CTCL to develop more effective treatments and improve patients’ survival [8].

The combinational use of multiple epigenetic modulators simultaneously or in conjunction with other treatments has been demonstrated to be effective in preclinical and clinical studies [9,10,11,12,13]. Combinational therapies that target several signaling pathways and clonal subpopulations can increase the survival and quality of life of patients [14]. An example of a combinational therapy involving existing drugs is the use of JAK/HDAC inhibitors, which have already been employed in haematological malignancies, presenting a prospective therapeutic target for CTCL. Recent NGS data, including single-cell sequencing, have uncovered genetic abnormalities in critical signaling networks and epigenetic components that play a significant role in CTCL pathogenesis [14]. High-throughput screening has become a popular tool to quickly identify and prioritize new medicinal molecules. Several studies on CTCL have used high-throughput technology, next-generation sequencing (NGS), and whole genome sequencing (WGS) techniques to detect potential key signaling pathways and targets [15,16,17]. Activation of the JAK-STAT pathway has been mostly associated with the pathogenesis and progression of CTCL, as well as some other hematologic malignancies [18,19]. Previous in vitro data from our group have shown that the combination of Resminostat (HDACi) with Ruxolitinib (JAKi) exhibited cytotoxic effects in CTCL cell lines, and inhibited cell proliferation, suggesting a strong synergy of the two drugs [20]. The drug combination inhibited phosphorylation of STAT3, AKT, ERK1/2, and JNK in MyLa cells, while it reduced activation of AKT and JNK in SeAx. However, there are few in vivo CTCL models to validate the potential therapeutic impact of new agents.

In the present study, we have initially investigated the preclinical experimental model by implanting MF/SS cells in the Chick Embryo Chorioallantoic Membrane (CAM) [21,22]. This experimental model has become an appealing tool for in vivo assays for drug testing, tumor growth, and metastasis [23]. The chick embryo is a naturally immunodeficient model and it can be used as a Sezary model to study the tumor microenvironment in CTCL during the early stages. The immune system of chicks does not begin to function until they are about 2 weeks old [24,25]. T cells appear at day 11 and B cells at day 12 [26], and chick embryos are immunocompetent by day 18 [24,25]. Chick CAM allows the fast vascularization of tumors placed on its surface. More specifically, CAM vasculature is attracted to grow into the developing tumor depending on the aggressiveness of the tumor cells, which then intravasate into the blood vessels. Unlike normal mouse models, most cancer cells arrested in the CAM microcirculation survive without causing cell injury, and a considerable number complete extravasation within 24 h following injection [27]. In comparison to mammalian models, where tumor growth takes 3 to 6 weeks, chick CAM is faster: microtumors appear 2 to 5 days after tumor cell transplantation. Finally, the model’s ease of use and low cost make it more appealing. Because of the short time (8–10 days) between implantation and chick hatching, most tumor cells are unable to develop macroscopic visible colonies in secondary organs in the CAM model [28].

Xenografted tumors from MyLa and SeAx cells implanted on the top of CAM were studied in order to assess CAM as a pre-clinical CTCL model for pharmacological testing. Based on our in vitro findings [20], we investigated the potential anti-tumor effects of JAKi/HDACi, Resminostat, and/or Ruxolitinib. Our data suggest that the CAM assay presents a promising CTCL pre-clinical model for testing future therapeutic agents.

## 2. Materials and Methods

### 2.1. Cell Lines and Culturing

The human CTCL cell lines, MyLa and SeAx, were kindly provided by Dr Michel Laurence (Skin Research Center Service de Dermatologie Hôpital Saint-Louis, INSERM, Paris, France), which were already tested and authenticated. Both cell lines were cultured in RPMI 1640, supplemented with 10% fetal bovine serum and 1% penicillin/streptomycin, at 37 °C in a humidified atmosphere with 5% CO_2_, for 24 h.

### 2.2. Drugs Tested

The HDAC inhibitor was a generous gift from 4SC AG (Planegg-Martinsried, Germany) and the JAK inhibitor was a gift from Novartis Incyte (Basel, Switzerland). Both inhibitors were dissolved in DMSO according to the manufacturers’ instructions. Therefore, vehicle controls or untreated cells were treated with 0.1% DMSO for all the experiments.

### 2.3. Chick Embryo CAM Model: Xenografted Tumours and Spontaneous Metastasis

The chick embryo CAM model was developed as previously described [21,22], by first preparing the eggs for xenografting tumor cells, then preparing tumor cells for grafting, grafting the tumor cells onto the CAM, and, finally, harvesting tumors and chick embryo tissues. More specifically, 10^6^ MyLa or SeAx cells were injected onto CAM embryos and induced tumor formation. Tumor growth was monitored, and spontaneous metastasis was initiated in chick embryos. An amount of 15 μΜ Ruxolitinib and 5 μΜ Resminostat were administered topically every two days. Harvesting was performed on day 7 and the numbers of tumor cells on the CAM, liver, and lung were analyzed by Alu PCR, along with their effects on cell viability and apoptosis.

The chick embryo CAM assay did not require administrative procedures for obtaining ethics committee approval for animal experimentation, since the chick embryo is not considered as a living animal until day 17 of development. The CAM was not innervated, and experiments were terminated before the development of centers in the brain associated with pain perception, making this a system not requiring animal experimentation permissions. All experiments were performed according to the national guidelines for animal care in accordance with the European Union Directive.

### 2.4. Quantitative Detection of Human Tumor Cell Metastasis

Genomic DNA was extracted from the harvested tissues using the Qiagen DNA purification system (Qiagen, Hilden, Germany catalog number: 158906;158910;158914). To detect human cells in the chick tissues, primers specific for the human Alu sequences (sense: 5′ ACGCCTGTAATCCCAGGACTT; 3′antisense: 5′ TCGCCCAGGCTGGCTGGGTGCA 3′) were used to amplify the human Alu repeats present in genomic DNA that was extracted from chick tissues. The real-time PCR used to amplify and detect Alu sequences contained 30 ng of genomic DNA, 2 mm MgCl_2_, 0.4 μm each primer, 200 μm DNTP, 0.4 units of Platinum Taq polymerase (Invitrogen Corporation, Carlsbad, CA, USA), and a 1:100,000 dilution of SYBR green dye (Molecular Probes, Eugene, OR, USA). Each PCR was performed in a final volume of 10 μL under 10 μL of mineral oil with the iCycler iQ (Bio-Rad laboratories, Hercules, CA, USA) under the following conditions: polymerase activation—95 °C for 2 min, 40 cycles at 95 °C for 30 s, 63 °C for 30 s, 72 °C for 30 s. A quantitative measure of amplifiable chick DNA was obtained through the amplification of the chick GAPDH genomic DNA sequence with chGAPDH primers (sense: 5′ GAGGAAAGGTCGCCTGGTGGATCG 3′; antisense: 5′ GGTGAGGACAAGCAGTGAGGA ACG 3′) using the same PCR conditions as described for Alu. The fluorescence emitted by the reporter dye was detected online in real-time, and the threshold cycle (Ct) of each sample was recorded as a quantitative measure of the amount of PCR product in the sample. The Ct is the fractional cycle number at which the fluorescence generated by the reporter dye exceeds a fixed level above baseline. When indicated, the Alu signal was normalized against the relative quantity of GAPDH and expressed as ΔCt = (CtGAPDH − Ct Alu). The changes in Alu signal relative to the total amount of genomic DNA (and, hence, changes in the quantity of human DNA in the chick tissue) were expressed as ΔΔCT = ΔCtcontrol − ΔCttreatment. Relative changes in metastasis were then calculated as 2ΔΔCT. Each assay included a negative control, a positive control, a no-template control, and the experimental samples in triplicate. To approximate the actual number of tumor cells present in each tissue sample, a standard curve was generated through quantitative amplification of genomic DNA extracted from a serial dilution of MyLa and SeAx cells respectively mixed with individual chick lung homogenates. By interpolating the Alu signal from experimental samples with the standard curve, the actual number of tumor cells/lung could be determined over a range of 50–100,000 cells/lung. Data processing and statistical analysis were performed using GraphPad Prism 9 (GraphPad Software Inc., San Diego, CA, USA) and Microsoft Excel 16.3 (Microsoft Corporation, Redmond, WA, USA). The synergistic effect between Resminostat and Ruxolitinib was determined by the combination index (CI) as previously described [29]. The CI value was determined by the following equation: CI = sum of tumor growth or metastasis inhibition of single agent treatment/tumor growth or metastasis inhibition upon combined treatment. A combination index (CI) of >1 indicates antagonism, a CI of 1 denotes additivity, and a CI of <1 indicates synergism. More specifically, CI values ranging from 0.1 to 0.3 are considered to indicate strong synergism, 0.3 to 0.7 synergism, and 0.7 to 0.85 moderate synergism.

### 2.5. Experimental Metastasis Chick Embryo Model

Fertilized chicken eggs were obtained from Gibert farm (Tarragona, Spain) and incubated with rotation at 37.50 °C and 60% humidity. On day 12 of incubation, the developing embryos were injected intravenously with 5 × 10^4^ cells in 0.1 mL serum-free DMEM. At indicated time points, the embryos also either received Resminostat, Ruxolitinib, both, or DMSO as a vehicle. On day 5, portions of the CAM were harvested to perform biochemical analyses and to determine, by Alu qPCR, the number of human tumor cells which had colonized the tissues.

### 2.6. Live Cell Imaging

MyLa and SeAx cells were labeled with 5 µmol/L green CellTracker CMFDA and injected intravenously at 1 × 10^5^ cells per embryo and the indicated concentrations of Resminostat and Ruxolitinib. To highlight vasculature, the embryos were injected with 50 µg of Rhodamine-conjugated Lens culinaris agglutinin (LCA, Vector, Burlingame, CA, USA). At 24 h, the embryos were sacrificed; the portions of the CAM were stretched on glass slides and examined in a Carl Zeiss Axio Imager microscope. Digital images were taken with AxioVision Rel. 4.6 software (Carl Zeiss MicroImaging).

### 2.7. Proliferation Analyses

Proliferation assays were performed using AlamarBlue Cell Viability Reagent (Thermo Fisher, Waltham, MA, USA). 5000 cells/well were plated in a 96 well plate in 100 μL medium and treated under the desired conditions. After that time, 10 μL of AlamarBlue Reagent was added and incubated in the dark at 37 °C for 12 h. Absorbance was read at 540 and 620 nm.

### 2.8. Apoptosis Assays

10^6^ cells were collected by centrifugation at 1000 rpm for 5 min at 4 °C. They were washed with 1 mL of filtrated 3 mM EDTA PBS and centrifuged again. The pellet was resuspended in 200 μL of binding buffer (BB) (10X BB: HEPES 0.1M pH 7.4, NaCl 1.4 M, CaCl^2^ 25 mM) and placed in cytometry tubes. Then, 1 μL of FITC Annexin V (BD Pharmagen) and 20 μL of FBS were added to avoid unspecific interactions. The mix was incubated 30 min in dark at 4 °C. After incubation, one wash with 1 mL of 2 mM EDTA PBS was performed, and the cells were collected by centrifugation and resuspended in 200 μL of 2 mM EDTA PBS to perform the flow cytometry. Apoptosis rate was determined in MACSQuant VYB (Miltenyi Biotec, Gladbach, Germany) and the results were analyzed with Flow Logic software (Miltenyi Biotec, Gladbach, Germany).

### 2.9. Migration Assays

Cell migration was examined in Transwell cell culture chamber filters (8 μm pore) (Corning, New York, NY, USA). Green tracker fluorescent-labeled cells were seeded at 5  ×  10^4^ cells in RPMI-0.2% FBS. Following 16 h incubation, the invading cells were fixed and analyzed by fluorescence microscopy and counted. Images were processed and analyzed using FIJI Image.

### 2.10. Invasion Assays

In Matrigel invasion assays, the upper sides of membranes (8 μm pore Transwell, Fisher Scientific, Hampton, NH, USA) were pre-coated with a dilution of 1:25 Matrigel (BD Biosciences, Franklin Lakes, NJ, USA) and 10% FBS-DMEM was added as a chemoattractant in the lower chamber. A number of 2.5 × 10^5^ MyLa and SeAx cells were plated in 150 μL of SF-DMEM in the upper chamber. Following 48 h incubation, the invaded cells were fixed and analyzed by fluorescence microscopy and counted. Images were processed and analyzed using FIJI ImageJ 2.3.1 (NIH, Bethesda, MD, USA).

### 2.11. 3D Tumour Spheroid Invasion Assay

MyLa and SeAx cells (4 × 10^3^ cells/well diluted in 200 µL) were seeded in low-adherence 96 well U-bottom plates (6055330, PerkinElmer, Waltham, MA, USA) and pelleted by centrifugation (100× *g* for 5 min). Two days after, tumor spheroid formation was visually confirmed, and the 3D invasion assay was performed. For that, 150 µL/well of growth medium was removed from the spheroid plates and 50 µL of BMM (356234, Corning, New York, NY, USA) was gently dispensed into the bottom well. To make sure the spheroids were in a central position, they were centrifuged (300× *g* for 3 min at 4 °C). Then, BMM was allowed to solidify for 1 h (37 °C) and 100 µL/well of complete growth medium including inhibitors (3× the desired final concentration) were added. Spheroids were monitored for 4 days by brightfield microscopy, imaging them each for 24 h. Then, images were analyzed using QuPath1 to quantify spheroids’ size and the number of invading cells.

### 2.12. Immunohistochemistry

After extraction, tissues were embedded in paraffin and cut using a microtome; 5 µm sections were then placed on poly-L-lysine-treated slides. Before staining, slides were deparaffinized, and tissues were rehydrated. After that, the slides were dried (1 h at 60 °C or overnight at 37 °C). Then, permeabilization of cells was carried out by incubation (10 min) of tissues with 0.1% IGEPAL in 1× Tris Buffered Saline (TBS). After that, the specimens were washed (2×, 5 min with 1× TBS), and non-specific bindings were blocked using a serum-free blocking agent, background punisher (BIOCARE Medical, Pacheco, CA, USA), for 10 min. Then, 1% Bovine Serum Albumin (BSA) 0,1% IGEPAL in 1× TBS solution with a primary mouse anti-rat CD44 antibody (diluted 1:100) (Antibodies Online, Aachen, Germany) or without primary antibody (negative control) was incubated overnight (4 °C). After incubation, specimens were washed in 1× TBS and incubated with 3× hydrogen peroxide in 1× TBS for 30 min to quench endogenous peroxidase. After that, the slides were washed again using 1× TBS and incubated (1 h) with a secondary anti-mouse biotinylated antibody (Vector Laboratories, Burlingame, CA, USA) (dilution 1:500 in 1% BSA 0.1% IGEPAL in 1× TBS). Then, slides were washed, as described above, and tissues were incubated (30 min) with horseradish peroxidase Avidin D (dilution 1:500 in 1× TBS). Finally, slides were washed again and incubated (5–10 min) with diaminobenzidine (Gibco, Gaithersburg, MD, USA). In addition, tissues were stained with hematoxylin, dehydrated, cleared, and mounted with DPX. Micrographs were captured by a Zeiss Axio Scope A1 microscope (Carl Zeiss, Oberkochen, Germany).

### 2.13. Western Blotting

CAM tumors were lysed with RIPA buffer (Sigma Aldrich, Burlington, MA, USA), supplemented with phosphatase and protease inhibitors (Roche, Basel, Switzerland). Whole cell lysates were subjected to acrylamide SDS-PAGE using standard procedures, transferred onto a nitrocellulose support membrane (Amersham Protran, GE Healthcare Life Science, Marlborough, Mass, United States), and Western blotted. All primary and secondary antibodies were diluted 1:1000 and 1:5000 respectively, unless otherwise stated. The following antibodies were used: anti-α-Tubulin (Santa Cruz Biotechnology, Dallas, TX, United States sc-23948), anti-phospho-ERK (Santa Cruz Biotechnology, sc-7383, RRID:AB_627545), anti-p44/42 MAPK (Erk1/2) (Cell Signaling Technology, Danvers, Mass, Unites States (137F5), Cat# 4695, RRID:AB_390779), phospho p38 (Santa Cruz biotechnology, sc-166182), p38 (Santa Cruz Biotechnology, sc-728, RRID:AB_632140), phospho-AKT (Ser473) Antibody (Cell Signaling Technology, Cat#9271), AKT Antibody (Cell Signaling Technology, Cat #9272), p-Stat5A/B (Santa Cruz Biotechnology, (5G4), sc-81524), Stat5 (Santa Cruz Biotechnology, (A-9), sc-74442), and α Tubulin (Santa Cruz Biotechnology, DM1A, sc-32293). Secondary antibodies: P/N 925-32212; RRID AB_2716622, P/N 925-32213; RRID AB_2715510, P/N 926-32213; RRID AB_621848. Signals were visualized and recorded with an Odyssey^®^ M Imaging System LICOR (Lincoln, NE, USA).

### 2.14. Statistical Analysis

To evaluate results and statistical significance of control and experimental groups, we used Graph Pad Prism software and the Student’s *t*-test or ANOVA analysis. Each global mean was compared using the two-tailed unpaired Student’s *t*-test with a statistical significance of *p* < 0.05 (95% confidence interval). *p* values: * < 0.05, ** < 0.005, *** < 0.001, **** < 0.0001

## 3. Results

### 3.1. Resminostat, Ruxolitinib, and Their Combination Inhibit Cell Proliferation and Induce Apoptosis in CTCL Cells

We initially analyzed the cell viability and apoptotic effects of Ruxolitinib and/or Resminostat on MyLa and SeAx cells. It was shown that monotherapies decreased cell viability and enhanced apoptosis when compared to vehicle control without, however, reaching statistical significance (Figure 1A,B). On the other hand, the combination of the drugs significantly showed a 62% reduction in the viability of MyLa and 60% reduction in SeAx cell lines (*p* < 0.001) when compared to controlled vehicle (Figure 1A,B). In addition, the drug combination enhanced apoptosis in MyLa by 2.85-fold and in SeAx cells by 3.7-fold when compared to untreated cells (MyLa, *p* < 0.005; SeAx, *p* < 0.001) (Figure 1C,D).

### 3.2. Combined Treatment of Resminostat and Ruxolitinib Impairs CTCL Tumorigenesis and Metastasis in Spontaneous Metastasis CAM Assay

We proceeded to investigate the inhibitory effects of Resminostat and/or Ruxolitinib on tumorigenesis and metastasis, in the chick embryo spontaneous metastasis model. Upon inoculation with 10^6^ MyLa or SeAx cells onto their CAM, chick embryos developed large primary tumors (100–200 mg) within 7 days. The monotherapy inhibited primary tumor formation and CAM intravasation in both MyLa and SeAx embryos, but only in SeAx embryos was the reduction in tumor formation and CAM intravasation statistically significant when compared to vehicle (*p* < 0.005). On the other hand, the combination of Resminostat with Ruxolitinib was more effective in inhibiting the primary tumor formation (31% in MyLa embryos, *p* < 0.01; 52% in SeAx embryos, *p* < 0.005) (Figure 2A) and blocked CAM intravasation (82% reduction in MyLa (*p* < 0.005) and 92% reduction in SeAx embryos (*p* < 0.005)) (Figure 2B). Moreover, we detected that, in the control MyLa and SeAx embryos, neoplastic cells intravasated the distal CAM and disseminated to internal organs such as the lung and liver, forming secondary metastatic foci (Figure 2C,D). As far as the metastasis is concerned, it was demonstrated that the monotherapies blocked liver metastasis in both MyLa (Resminostat, *p* < 0.01; Ruxolitinib, *p* <0.005) and SeAx embryos (Resminostat, *p* < 0.005).

Lung metastasis was also blocked-in monotherapies (MyLa, Resminostat, *p* < 0.05; SeAx, Resminostat *p* < 0.005; Ruxolitinib, *p* < 0.005). As expected, the combination of drugs significantly blocked liver (76% reduction in MyLa, *p* < 0.001; 75% reduction in SeAx, *p* < 0.005) and lung metastasis (87% reduction in MyLa, *p* < 0.001; 79% reduction in SeAx, *p* < 0.001).

To further demonstrate that the combination of Resminostat and Ruxolitinib impairs CTCL tumor formation, we performed an immunohistological analyses. We observed a formed primary tumor in CAM, and its margin can be demarcated as shown in Appendix A. In addition, we found that the combination of drugs significantly blocked intravasated CTCL cells in CAM blood vessels.

Then, we determined the synergistic effect between Resminostat and Ruxolitinib by the combination index (CI) as previously described [29]. The combination treatment exhibited synergistic effects in tumor growth and metastasis inhibition in both cell lines. In MyLa cells, CI was 0.782, whereas in SeAx cells, the CI was 0.741. These values indicate moderate synergism in tumor growth inhibition for both cell lines. For CAM intravasation, MyLa cells’ CI was 0.194, whereas in SeAx cells, the CI was 0.213. For lung metastasis, MyLa cells’ CI was 0.261, whereas in SeAx cells, the CI was 0.282. For liver metastasis, MyLa cells’ CI was 0.279, whereas in SeAx cells, the CI was 0.295. These values indicate strong synergism in metastasis inhibition for both cell lines.

### 3.3. Combined Treatment of Resminostat and Ruxolitinib Impairs CTCL Migration and Invasion

Following the effect of Resminostat and Ruxolitinib in tumor formation and metastasis in vivo, we next investigated their effects on cell migratory and invasive potential. As it is shown in Figure 3, the migration trans-well assay was used, showing that monotherapies significantly reduced (*p* < 0.05) the migration of SeAx cells (Figure 3B,D). However, the combination of Resminostat with Ruxolitinb had a dramatic decrease in both cell lines when compared to untreated cells (MyLa—91% reduction, *p* < 0.005: SeAx—92% reduction, *p* < 0.005). On the contrary, in MyLa cells, only Resminostat resulted in a significant decrease (*p* < 0.05), as well as the combination therapy (*p* < 0.005), in migration. The invasion trans-well assay demonstrated that at 48 h post-treatment in MyLa cells, the combination of HDACi with JAKi was more effective (*p* < 0.005) than HDACi treatment alone (*p* < 0.05) (Figure 4A,C). On the other hand, in SeAx cells, the combination of the drugs was more effective (*p* < 0.005) when compared to both monotherapies alone (*p* < 0.05) (Figure 4B,D). Additionally, a 3D tumor spheroid invasion assay showed that the combination of Resminostat with Ruxolitinb inhibited the invasive ability of MyLa (*p* < 0.005) and SeAx cells (*p* < 0.005) (Figure 5). Quantitative analysis indicated a 40–50% reduction both in productive distance travelled by escaped CTCL cells and in the number of cells that had escaped from spheroids treated with Resminostat and Ruxolitinib.

### 3.4. Combined Treatment of Resminostat and Ruxolitinib Reduce CTCL Extravasation CAM Assay

We next validated the effect of combined treatment of Resminostat and Ruxolitinib during metastatic dissemination of CTCL cells. For this purpose, we analyzed the invasive ability of CTCL cells and the colonization behavior of MyLa and SeAx using the experimental metastasis model in chick embryo. The 3D invasion assay showed that the combined treatment inhibited the invasive ability of CTCL cells (MyLa—49% reduction distance, *p* < 0.005; SeAx—36% reduction in distance, *p* < 0.005) (Figure 5).

Interestingly, by day 5, the combination of Resminostat and Ruxolitinib produced a significant decrease in the number of human tumor cells detected in the CAM (MyLa—72% reduction, *p* < 0.005; SeAx—79%, *p* < 0.005), liver (MyLa—78% reduction, *p* < 0.005; SeAx—92% reduction, *p* < 0.001), and lung (MyLa—68% reduction, *p* < 0.005; SeAx—89% reduction, *p* < 0.001). Under these in vivo conditions, combined treatment reduced the colonization capability of CTCL cells (Figure 6). Then, we analyzed the colonization behavior of Myla and SeAx cells by live-cell imaging of fluorescently labeled tumor cells in the CAM tissue. Twenty-four hours after cell inoculations, control MyLa and SeAx cells appeared to extravasate normally from the terminal CAM capillaries. In contrast, the combination of Resminostat with Ruxolitinb dramatically decreased the number of extravasated cells (Figure 6D).

### 3.5. Resminostat, Ruxolitinib, and Their Combination Inhibit Key Signaling Pathways in CTCL Xenografted Tumors

To further investigate the signal transduction pathways that are activated by single or combinational Resminostat and Ruxolitinib treatment, we performed Western blot analyses in CAM tumors for several key implicated molecules. We investigated the expression of total p38 with the phosphorylated p-p38 (Tyr182), the total AKT protein with the phosphorylated p-AKT (Ser473), the total protein ERK with the phosphorylated p-ERK (Tyr204), as well as the total STAT5 with the phosphorylated p-STAT5 (Tyr694/699). Normalization of protein levels was achieved using the expression levels of α-tubulin. As is depicted on Figure 7 (Appendix A), in MyLa cells, the monotherapies resulted in a significant decrease only in the phosphorylation of p-AKT (*p* < 0.005). In contrast, the combination of Resminostat with Ruxolitinib inhibited the phosphorylation of p-p38 by 58% (*p* < 0.001), p-AKT by 64% (*p* < 0.0001), p-ERK by 83% (*p* < 0.0001), and p-STAT5 by 45% (*p* < 0.005). In SeAx cells, the monotherapies showed a significant inhibition in the phosphorylation of p-ERK (*p* < 0.005), whereas the combinational treatment was more effective in the reduction of the phosphorylation of p-p38 by 35% (*p* < 0.005), p-AKT by 37% (*p* < 0.05), p-ERK by 93% (*p* < 0.0001), and p-STAT5 by 30% (*p* < 0.005).

## 4. Discussion

Currently, MF/SS are incurable diseases with a dismal prognosis in advanced stages and urgent requirement of more effective treatment. The selection of therapy is mainly based on disease stage. Though more effective treatments are available for early-stage disease, late-stage disease treatments remain largely ineffective [30]. In selected MF/SS cases, stem cell transplantation presents the only treatment option [31,32]. Brentuximab and Mogalizumab are ‘milestone’ additions to CTCL therapy. Combinational treatments produce higher complete response rates, however disease-free survival and overall survival do not differ from sequential conservative therapy [33,34,35,36,37,38,39,40,41,42]. New or repurposed drugs for CTCL need to be tested in a well-established, functional, and reproducible in vivo model. However, a representative in vivo CTCL model is currently lacking to study the development and progression of MF/SS, and is highly demanded for the validation of in vitro results and detection of promising therapeutics that may lead to clinical trials. The present study aimed to establish an in vivo model for MF/SS, characterized by rapid tumor growth after transplantation of an unselected ‘primary’ malignant T cell line, with a high, robust, and predictable tumor take, and finally, the creation of macroscopically visible secondary tumors. To the best of our knowledge, this is the first study that developed and employed an MF/SS chicken CAM model for the evaluation of new combinational treatment options in CTCL in vivo.

Although a few preclinical in vivo CTCL models have been previously developed, they present several problems, mainly attributed to the small availability of CTCL cell lines, which are derived from blood samples of patients with CTCL, the difficulties to grow them in vitro [43,44,45], and the inability to obtain tumor metastasis macroscopically [46]. Current animal models of CTCL include transplantation of human skin grafts into mice with severe combined immune insufficiency (SCID), or subcutaneous transplantation of MyLa cells to the flanks of athymic nude mice lacking T cells and T cell-dependent antibody responses, to generate an alternative mouse xenograft model. Although a subcutaneous site improves tumor growth monitoring, it is neither a physiological nor an optimal environment for CTCL cell line expansion. In fact, most studies in immunodeficient mice involving subcutaneous injection of fresh cells from patients or CTCL cell lines have failed. In addition, CTCL cells may need an appropriate homing site delivering the necessary growth factors, and, thus, studies have considered the intrahepatic mouse route as interesting for CTCL, in which malignant T cells often circulate [47]. A mouse model for both aggressive and indolent CTCL cell line engraftment has been established to evaluate differences in CTCL tumorigenicity, migration, and spreading capacities, but further studies are needed to evaluate its functionality and reproducibility [48].

Unlike mouse models that have been previously employed in CTCL research, the chick embryo CAM assay (in ovo and ex ovo) is a simple, easy, and fast in vivo model where microtumors appear in 2 to 5 days after tumor cell transplantation, compared to 3 to 6 weeks in mammalian models. The benefits of the CAM assay include easy monitoring of the tumor extravasation and intravasation into the microvasculature and consequent formation of metastases in organs, evaluated by qPCR [49,50]. Regarding drug testing, most research has used the CAM assay to investigate the antiangiogenic or angiogenic potential of compounds/materials and their impact on embryo development [51,52]. The CAM assay can be used to study the activity of candidate molecules (new or repurposed) for anticancer therapy in vivo without the need for a rodent facility or ethical approvals for animal experimentation, since the CAM is not innervated, and tests are not connected with pain perception by the embryo. Current animal experimentation legislation in the European Union and Switzerland enable testing with chick embryos without the approval of animal experimentation committees.

Taking into account the tremendous advantages of the chick embryo CAM model, we proceeded to expand our previous in vitro effects of Resminostat and Ruxolitinib combination in vivo [20].

HDAC inhibitors are a promising class of therapeutic agents for a wide range of cancers [53,54]. It is known that HDAC inhibitors that are currently in use for CTCL are Vorinostat (FDA approved, 2006) and Romidepsin (FDA approved, 2009). On the other hand, Resminostat is under approval, since it is running the pivotal European RESMAIN study, aiming for the evaluation of Resminostat in maintenance treatment of patients with advanced stage (Stage IIB-IVB) MF/SS that have achieved disease control with systemic therapy. Despite the approval by the FDA for the treatment of certain cancers, HDACi have been shown to have a limited therapeutic efficacy against solid tumors as a single therapeutic agent. HDAC inhibitors reduce JAK-2 expression, likely due to effects on JAK-2 mRNA expression and through increased JAK-2 proteasomal deterioration [55,56,57]. The curative potential of JAK inhibitors appears to be limited, and the survival benefits are controversial with limited follow-up available [58]. However, HDACi have been shown to function synergistically with a range of structurally and functionally diverse chemical compounds, biologically active polypeptides, and novel immune therapies. Combining HDACi with other cancer therapeutics may thus be an avenue to achieve their full therapeutic potential [59,60,61,62]. 

Ruxolitinib is an oral JAK1/2 inhibitor, recently FDA-approved for treatment use in myelofibrosis and polycythemia Vera. It is also indicated for the treatment of various solid tumors (breast, pancreatic, colorectal, head and neck, and prostate) and hematologic illnesses (CLL, ALL, AML, CML, and NSCLC). Findings from pooled 5-year data from COMFORT trials [63] demonstrated long-term OS benefit from Ruxolitinib, but Ruxolitinib resistance also develops following chronic drug exposure [64], highlighting a clear need for combined therapies; it would be advantageous to conduct a clinical trial in the near future to overcome the resistance and toxicity of monotherapy in CTCL patients. Combination therapy might prove beneficial due to synergistic impacts on oncogenic transformation, which can enable the effective use of lower doses of the different agents with better tolerability, and might avoid or delay the development of drug resistance, as is shown in other hematological malignancies.

The combination of Resminostat, an HDACi, with Ruxolitinib, a JAKi exhibiting synergistic antitumor effects, resulted in the blockade of metastasis and inhibition of key molecules in implicated signaling pathways in the chick embryo xenograft model of CTCL. We observed that a single administration of Resminostat or Ruxolitinib inhibited cell viability and induced apoptosis of both CTCL cell lines.

Monotherapies decreased tumor formation and blocked CAM intravasation in spontaneous metastasis assay, but only in SeAx cells was the change statistically significant, indicating a differential response of SS from MF. As far as the effect of monotherapies in migration and invasion were concerned, we demonstrated that in MyLa cells, only the Resminostat significantly decreased migration and invasion, whereas in SeAx cells, both monotherapies were effective. This finding further demonstrates that different mechanisms are implicated during CTCL progression. These differences could be explained by the heterogeneity in cell origin and CTCL subtype between MF and SS.

Interestingly, the drug combination was most effective in inhibiting cell proliferation, inducing apoptosis, decreasing tumor formation, and blocking CAM intravasation. These altered cellular functions were mainly attributed to the drug synergy which induced a strong inhibition of key signaling pathways such as AKT, MAPK, and JAK/STAT. It has been shown that Ruxolitinib inhibits cell proliferation, STAT activation, and DNA synthesis, while activating apoptosis in CTCL cell lines [20,65,66]. On the other hand, Resminostat has been shown to restrain the phosphorylation of 4EBP1 and p70S6K, indicating a deregulation in AKT signaling [67]. Treatment of CTCL cell lines with Resminostat was demonstrated to reduce Bim and Bax protein levels along with Bcl-xL [67]. The study of Yumeen et al. (2020) supported the clinical implementation of Ruxolitinib as a novel therapy for leukemic CTCL and further enforced the synergistic potential combination of Ruxolitinib with BCL2, HDAC, BET, or proteasome inhibition [68]. On the other hand, HDAC inhibitors were shown to reduce JAK-2 expression, possibly due to effects on JAK-2 mRNA expression and through increased JAK-2 proteasomal deterioration [66]. Particularly, Vorinostat, another HDACi, and Ruxolitinib together enhanced STAT5 dephosphorylation, inhibiting its pathway [66]. Civallero et al. demonstrated that the combination of Ruxolitinib with Vorinostat in CTCL could affect cell proliferation by targeting the glycolytic and oxidative pathways [66].

In order to further elucidate the mechanism of action of the proposed synergistic treatment, additional functional investigation is required to elucidate the underlying mechanism of action since it is a multifactorial event which is determined by the extensive crosstalk between the different signal transduction pathways implicated in CTCL. When the effect of Resminostat or Ruxolitinib was investigated on CAM tumors, those arisen by MyLa cells showed a significant decrease in the phosphorylation of AKT when compared to the untreated CAM tumors. On the contrary, CAM tumors arisen by SeAx cells showed a dramatic inhibition in the phosphorylation of ERK when treated with either Resminostat or Ruxolitinib. We have previously shown that activation of the AKT/mTOR pathway in MF is correlated with NOTCH1, p-ERK, and p-STAT3, and is implicated in the acquisition of a more aggressive phenotype. Moreover, the combination of p-AKT, p-p70S6K, and p-4E-BP1 emerged as a significant potential prognostic marker in patients with advanced disease stage [69,70]. Drug combination also blocked the signaling of p-p38, p-AKT, p-ERK, and p-STAT in MyLa cells, and p-ERK and p-STAT in SeAx cells. This could be attributed to the drug synergy that we showed from the combination of Resminostat with Ruxolitinib. On the contrary, our in vitro data previously demonstrated that the Resminostat/Ruxolitinib drug combination affected the activation of AKT in both cell lines, whereas it also inhibited JAK/STAT and MAPK activation in the MF cell line. This finding further indicates the differential genetic and epigenetic mechanisms implicated in MF and SS, as well as the differences in signaling pathways in vitro and in vivo, since malignant T cells present variations in the activation of cellular signaling due to extensive crosstalk between different signal transduction pathways.

Our results revealed that the two drugs exhibit differential profiles of inhibition in terms of key signaling molecule activation in the CTCL cell lines tested both in vitro [18], as well as in the present study in vivo, further confirming that MF and SS should be considered as different diseases, arising from distinct T cell subsets [71,72,73]

Combination therapy of Resminostat and Ruxolitinib might prove beneficial for CTCL patients due to the synergistic impacts on basic cellular functions and on the inhibitory effect on key signaling molecules. This synergistic treatment could enable the effective use of lower doses of the different agents with better tolerability and might avoid or delay the development of drug resistance.

## 5. Conclusions—Future Directions

Our findings, using the chick embryo CAM spontaneous metastasis model, indicated that the JAKi/HDACi combination exhibited synergistic antitumoral effects and blocked CAM intravasation, as well as liver and lung metastasis, while it inhibited migration and invasion. The proposed drug combination also inhibited key signaling molecules, highlighting the significance of these pathways in the CTCL development and progression. Therefore, it may represent a promising novel therapeutic modality for CTCL patients. Importantly, the in vivo chick CAM metastasis model could be a good CTCL pre-clinical model to discover new treatments and further improve CTCL patients’ survival, who fail to benefit from monotherapy.

## Figures and Tables

**Figure 1 cancers-14-01070-f001:**
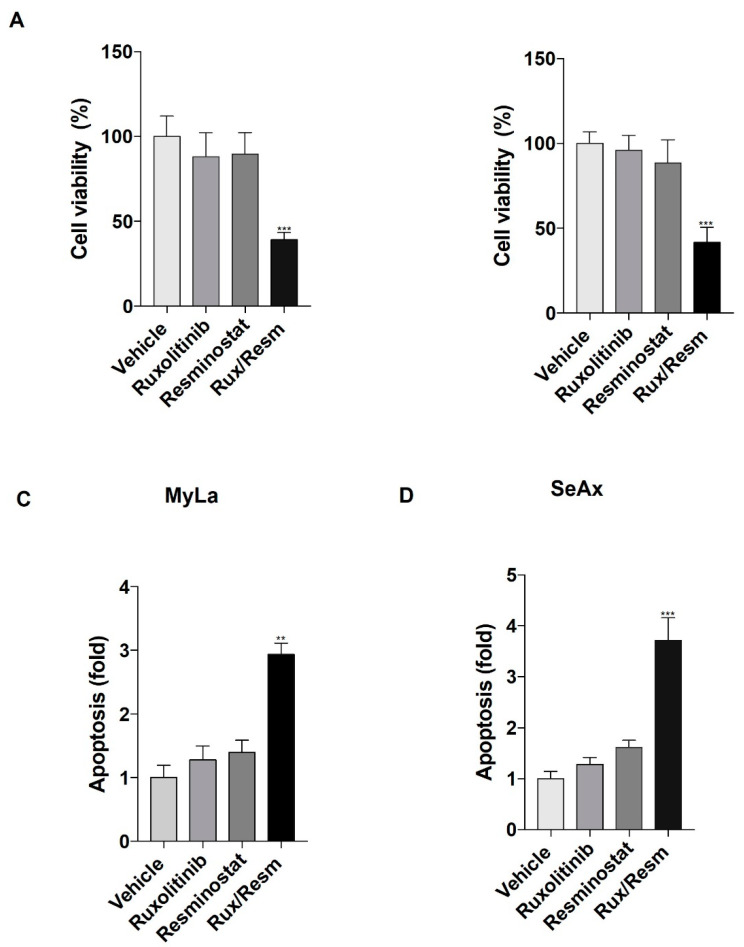
The effect of Resminostat, Ruxolitinib, and their combination in cell proliferation (**A**,**B**) and apoptosis (**C**,**D**) in CTCL cells. Cell proliferation was evaluated by AlamarBlue and apoptosis assay was performed by flow cytometry using FITC Annexin V. Data show mean ± SEM from three (*n* = 3) independent experiments. *p* values: ** < 0.005, *** < 0.001.

**Figure 2 cancers-14-01070-f002:**
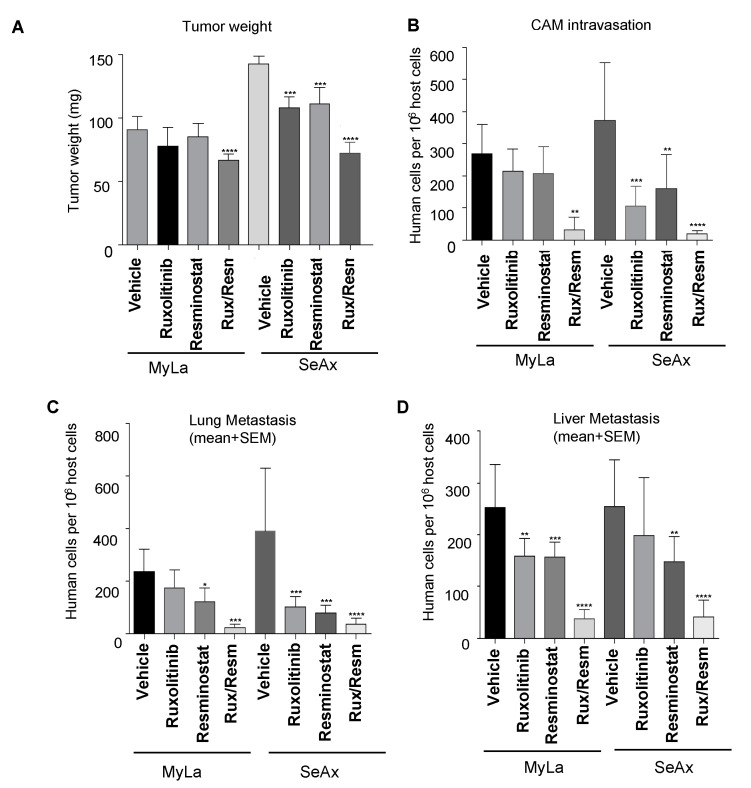
Evaluation of efficiency of Resminostat (Resm) and Ruxolitinib (Rux) to inhibit tumor growth (**A**), and metastatic spread of CTCL cells to chick chorioallantoic membrane (CAM) intravasation (**B**) and distant organs, lung and liver (**C**,**D**). CTCL cells were grafted onto the CAM of chick embryos (1 × 10^6^ cells per embryo). Developing tumors were treated on days 2 and 4 with topical applications of corresponding agents. On day 7, the levels of tumor cell intravasation to the CAM (**B**) and metastasis to distant organs were quantified by Alu PCR. Primary tumors were excised and weighed to determine the effect of the treatments on tumor growth. Data show mean ± SEM from three (*n* = 3) independent experiments, each employing from 14-18 embryos per treatment variant. *p* values: * < 0.05, ** < 0.005, *** < 0.001, **** < 0.0001.

**Figure 3 cancers-14-01070-f003:**
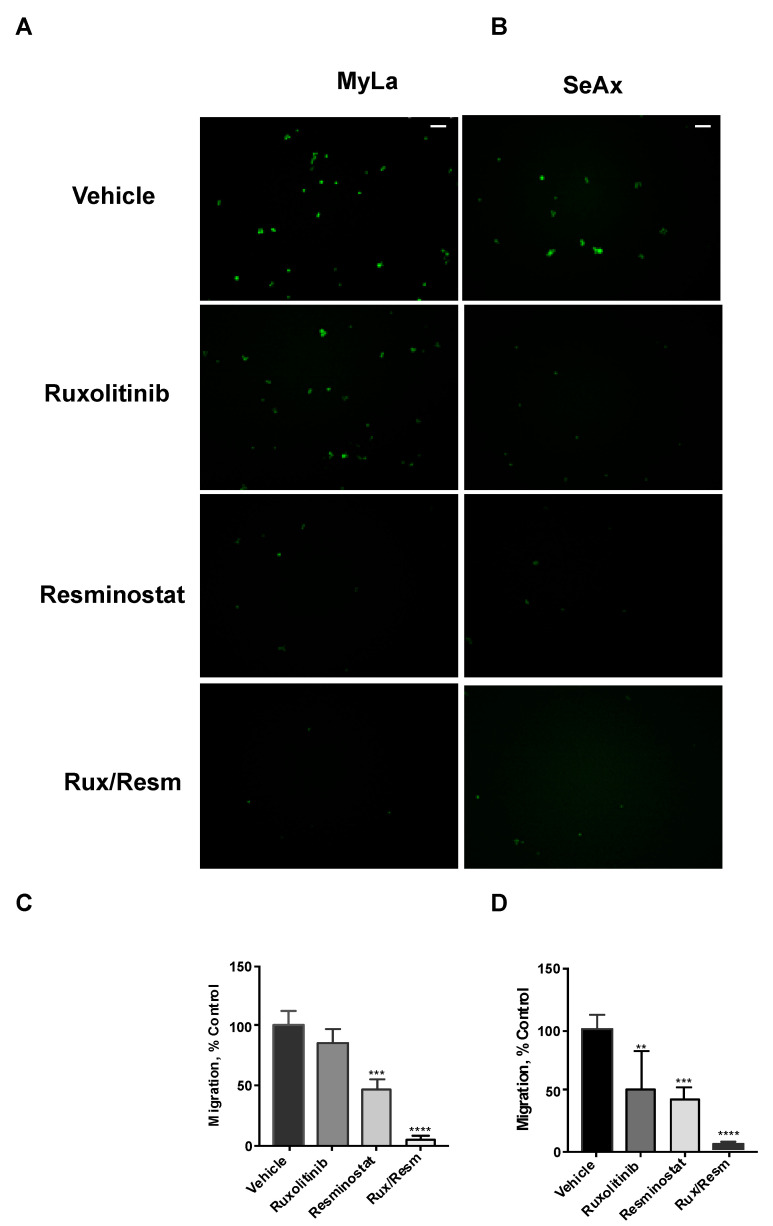
The inhibitory effect of combined treatment of Resminostat and Ruxolitinib in CTCL transwell migration. Fluorescent-labeled cells were placed into trans-well inserts. Tumor cells that crossed the pores’ membrane were quantified after 16 h. Representative pictures of migration assay using Myla (**A**) and SeAx (**B**) cells treated as indicated. Quantification of invasion of Myla (**C**) and SeAx (**D**) cells. Scale bar, 100 μm. Images from a representative experiment out of three independent experiments were performed in triplicate. Data are mean ± SEM. *p* values: ** < 0.005, *** < 0.001, **** < 0.0001.

**Figure 4 cancers-14-01070-f004:**
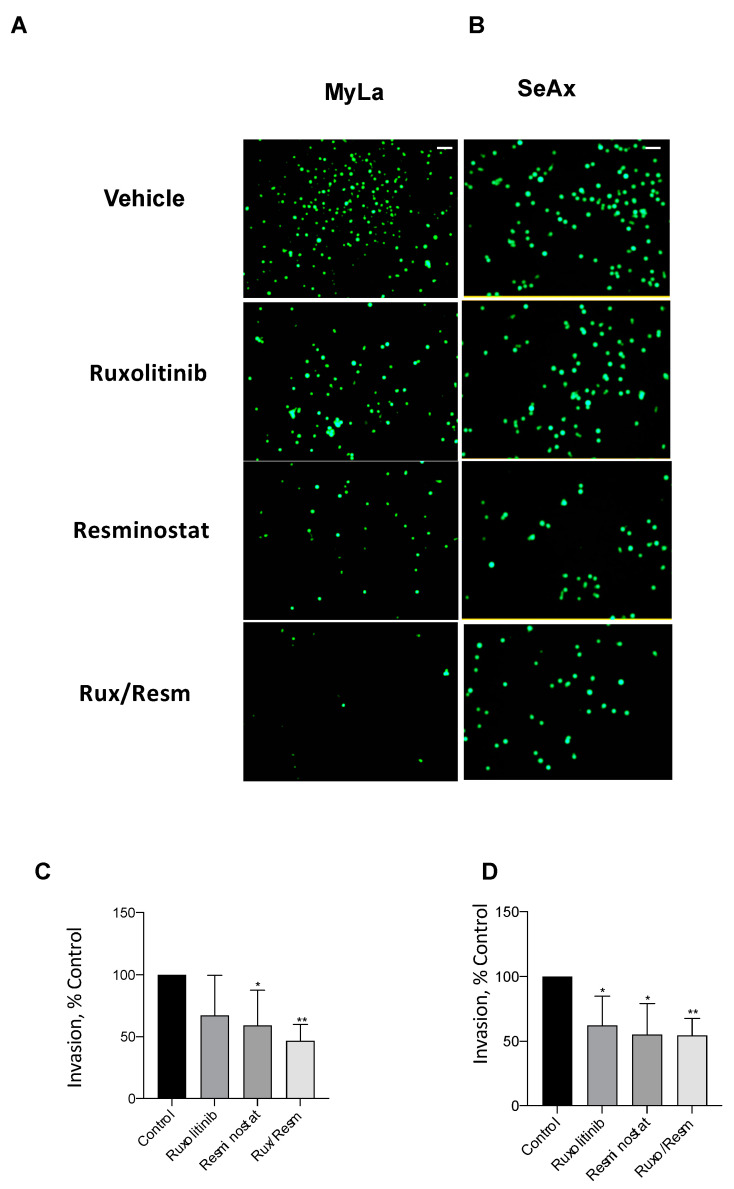
The inhibitory effect of combined treatment of Resminostat and Ruxolitinib in CTCL transwell invasion assay. Fluorescent-labeled cells were placed into Matrigel-covered inserts. Tumor cells that crossed Matrigel and pores of the membrane were analyzed after 48 h by fluorescent microscopy. Representative pictures of transwell invasion assay using Myla (**A**) and SeAx (**B**) cells treated as indicated. Quantification of invasion of Myla (**C**) and SeAx (**D**) cells. Scale bar, 100 μm (Data show mean ± SEM from three (*n* = 3) independent experiments performed in triplicate. *p* values: * < 0.05, ** < 0.005.

**Figure 5 cancers-14-01070-f005:**
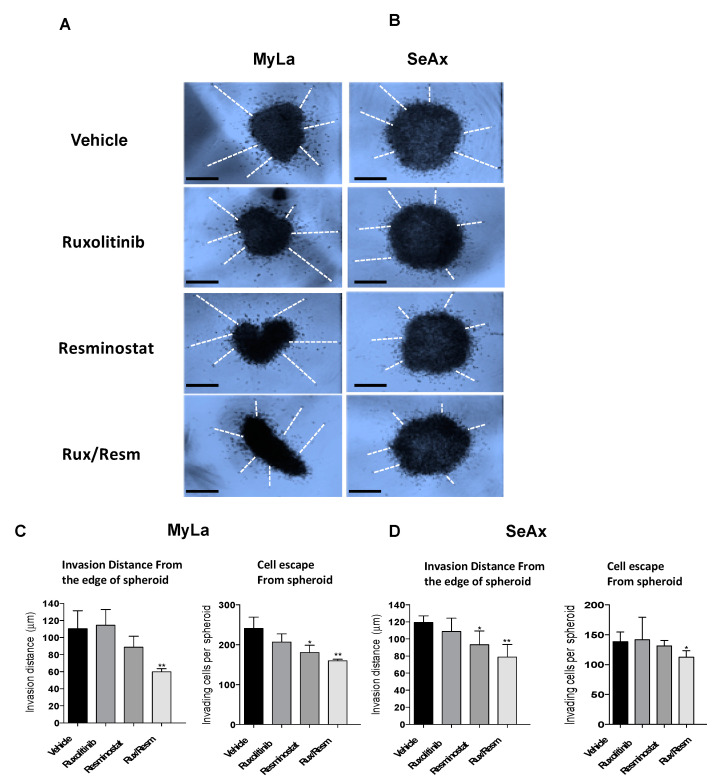
3D tumor spheroid invasion assay to determine the efficiency of the combination of Resminostat with Ruxolitinib. Representative pictures of Myla (**A**) and SeAx (**B**) 3D spheroids after 48 h. The combined therapy inhibited the invasive ability of CTCL cells. Representative images of CTCL spheroids showing cell escape and invasion. (**C**,**D**) Quantification of spheroid cell escape and invasion. Mean invasion distance covered by escaped cells from the edge of the spheroid and (**left** graphs) and the number of cells escaped from the spheroid (**right** graphs) were determined in acquired images after 48 h. Images from a representative experiment and data show mean ± SEM from three (*n* = 3) independent experiments performed in triplicate. Scale bar: 50 μm *p* values: * < 0.05, ** < 0.005.

**Figure 6 cancers-14-01070-f006:**
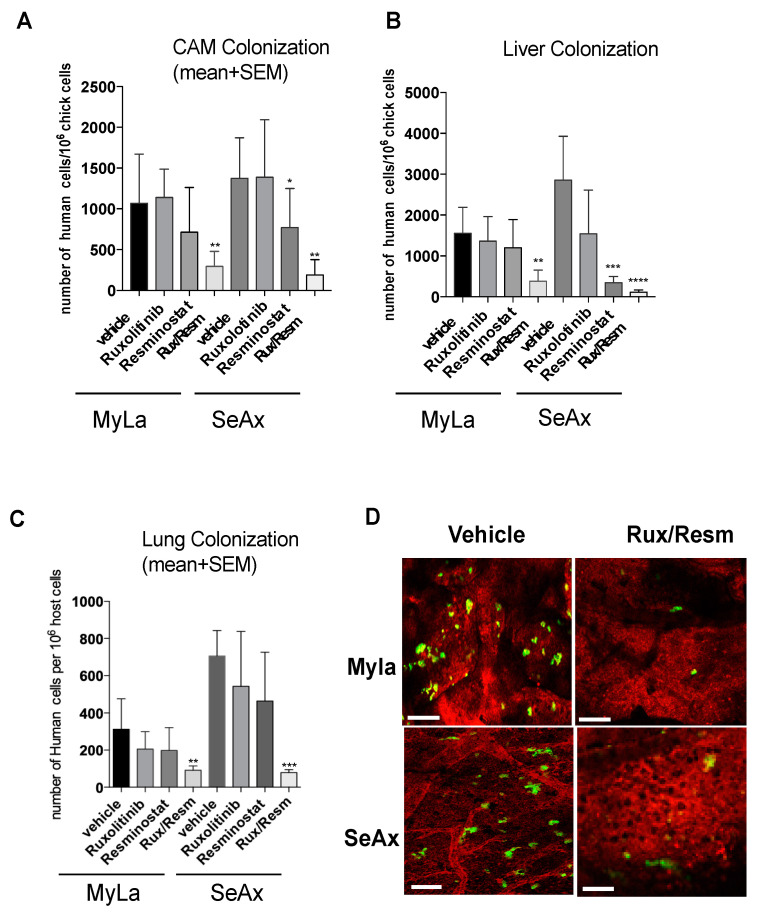
The effect of Resminostat and Ruxolitinib during colonization of CTCL cells in an experimental metastasis model in chick embryo. The combination of Resminostat and Ruxolitinib produced a significant decrease in the number of human tumor cells detected in CAM (**A**), liver (**B**), and lung (**C**). Chick embryos were injected with Myla or SeAx cells. At day 5, the levels of colonization in organs were analyzed by Alu PCR. The bars are the means determined in three (*n* = 3) independent experiments using from 12–16 embryos per variant. *p* values: * < 0.05, ** < 0.005, *** < 0.001, **** < 0.0001. (**D**). The colonization behavior of MyLa and SeAx cells by live-cell imaging of fluorescently labeled tumor cells in the CAM tissue. Live image analysis of CTCL cells was performed 24 h after green fluorescent-labeled cells were inoculated into chick embryos, vasculature was highlighted with red fluorescent Rhodamine. Scale bar: 50 μm.

**Figure 7 cancers-14-01070-f007:**
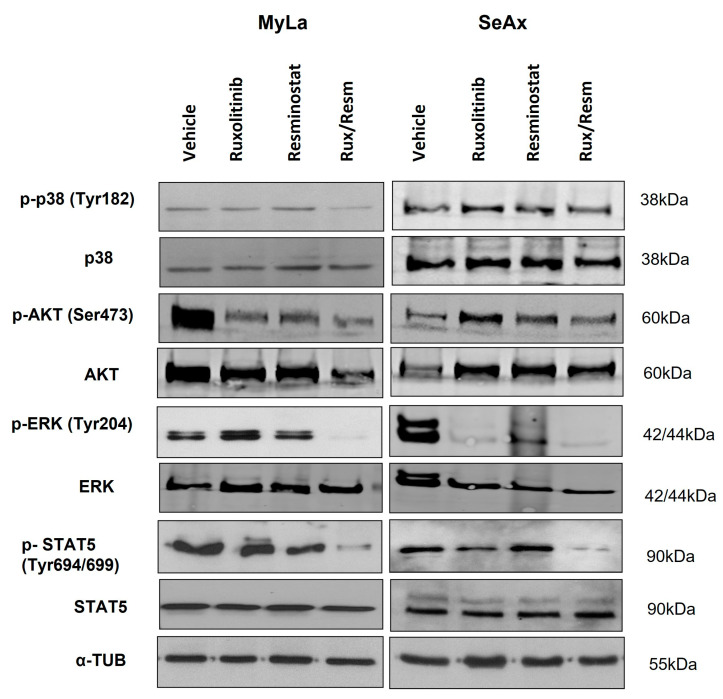
Western blot analyses in CTCL CAM tumors for key implicated pathways. Portions of primary tumors were lysed and analyzed for the activation levels of p-38 (Tyr1 82), p-AKT (Ser473), p-ERK (Tyr 204), p-STAT5 (Tyr694/699), and α-tubulin. Uncropped full western blot figures are included as Appendix A.

## Data Availability

The data presented in this study are available in this article (and Appendix A).

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
