# Peer review of "Combination of Resminostat with Ruxolitinib Exerts Antitumor Effects in the Chick Embryo Chorioallantoic Membrane Model for Cutaneous T Cell Lymphoma"

_cancers, 2022, doi:10.3390/cancers14041070_

Round 1

Reviewer 1 Report

Interesting, well written paper, but might be difficult to understand for those who are new to the topic.

My recommendation: abstract and conclusion section need minor corrections.

Author Response

Response to Reviewer 1 Comments

Point 1: Interesting, well written paper, but might be difficult to understand for those who are new to the topic. General comments:

-    Writing is adequate and no prove or language checks are required

-    Figure legends are presented with adequate detail

-    Please make sure that all abbreviations are always spelled the same

My recommendation: abstract and conclusion section need minor corrections.

Response: We would like to thank the reviewer for the recommendations. Both abstract and conclusion section were revised. We have also checked and corrected all the abbreviations used.

Reviewer 2 Report

I am pleased to review “Combination of Resminostat with Ruxolitinib exerts antitumor 2 effects in the chick embryo chorioallantoic membrane model 3 for Cutaneous T-cell Lymphoma “ (cancers-1591577).

Here, Karagianni et al. present a study building on prior work of the group to establish a CTCL pre-clinical model and investigate the combination of Resminostat and Ruxolitinib for the treatment of CTCL.

Overall, the manuscript offers high quality and is nicely written.

I only have some minor comments/questions.

Abstract: Offers a good summery of the manuscript and its major findings.

Introduction: Nicely written, presents key points necessary to understand the manuscript. Please add an additional sentence on current state of CTCL treatment options and their difficulties, which is the basis for this work.

Methods and Results: The methodology and experimental design are presented in adequate detail.

Going forward more experiments on the influence of the drugs (Resminostat & Ruxolitinib, each alone and especially in combination) on CTCL tumor cell would be desirable (phenotype changes etc.).

Discussion:  Discusses the findings adequately, while also mentioning current CTCL animal models. For a clearer and more complete picture please add some arguments on how this work rates compared to other work done to treat CTCL (and why Resminostat/Ruxolitinib might be a superior approach), alongside incorporating key difficulties in current CTCL therapy, which make the work in this manuscript and further research in this direction necessary.

Figures: Overall, figures are well presented. However, please make sure figure sizes (e.g. to have all significances (*) precisely over the bars) are similar in the different respective figures

General comments:

-    Writing is adequate and no prove or language checks are required

-    Figure legends are presented with adequate detail

-    Please make sure that all abbreviations are always spelled the same

Author Response

Response to Reviewer 2 Comments

I am pleased to review “Combination of Resminostat with Ruxolitinib exerts antitumor 2 effects in the chick embryo chorioallantoic membrane model 3 for Cutaneous T-cell Lymphoma “ (cancers-1591577). Here, Karagianni et al. present a study building on prior work of the group to establish a CTCL pre-clinical model and investigate the combination of Resminostat and Ruxolitinib for the treatment of CTCL. Overall, the manuscript offers high quality and is nicely written. I only have some minor comments/questions.

Abstract: Offers a good summery of the manuscript and its major findings.

Point 1: Introduction: Nicely written, presents key points necessary to understand the manuscript. Please add an additional sentence on current state of CTCL treatment options and their difficulties, which is the basis for this work.

Response: We would like to thank the reviewer for the nice comment regarding the introduction. We have added a short paragraph on the current state of CTCL treatment options and their difficulties.

Point 2:Methods and Results: The methodology and experimental design are presented in adequate detail. Going forward more experiments on the influence of the drugs (Resminostat & Ruxolitinib, each alone and especially in combination) on CTCL tumor cell would be desirable (phenotype changes etc.)

Response: The establishment of clinically relevant models for tumor growth, metastasis and drug testing is a major challenge in CTCL research. Here we report a relevant assay enabling quantitative analysis of metastatic capacity of CTCL tumor cells following implantation into the chorioallantoic membrane(CAM) of chick embryos. CAM model provides a platform for phenotypic and functional characterization of CTCL  metastatic cells and therapeutic compound screening approaches.

Malignant T cells in CTCL mostly develop from CD4 fraction and present the skin-tropic memory phenotype. Significant research efforts are directed towards uncovering the origin of T-cell transformation which is the major a hallmark of CTCL. Malignant T cells display phenotypic plasticity attributes of T-regulatory cells, which lead to complex heterogeneity of the malignancy. We have developed a model to get CTCL lesions on the CAM of chick embryos and determine tumor growth and metastasis. We plan to study the effect of the drugs (Resminostat & Ruxolitinib, each alone and in combination) in CTCL phenotypic changes during tumor progression, tumor microenvironment, immune response as well their capacity to induce angiogenesis. By mimicking repetitive metastatic processes on a chick embryo CAM) model, we will study CTCL metastatic cells  phenotypic changes . We will perform immunophenotypic analysis of CTCL metastatic cells obtained from blood and distant organs of chick embryos.  Moreover, we will determine the effect of Resminostat & Ruxolitinib in CTCL phenotypic changes during  the metastatic cascade.

Moreover, using the chick embryo spontaneous metastasis model (day 18, once the chick embryo immune system is developed) we will analyze immunological effectors that control the T-cell lineage differentiation in CTCL, which shifts towards T helper  and T regulatory cells. We will determine the effect of Resminostat & Ruxolitinib, each alone and especially in combination on Interleukin IL-2 and IL-15 levels. The role of immune effector cells in CTCL tumor is not clear, although changes in the composition of malignant T cells seem to occur during the progression of the malignancy. Thus we propose to analyze phenotypic changes and immune response “in vivo” during CTCL tumor progression. Our results will potentially improve prognosis and lead to more effective and targeted therapies for CTCL patients.

Point 3: Discussion:  Discusses the findings adequately, while also mentioning current CTCL animal models. For a clearer and more complete picture please add some arguments on how this work rates compared to other work done to treat CTCL (and why Resminostat/Ruxolitinib might be a superior approach), alongside incorporating key difficulties in current CTCL therapy, which make the work in this manuscript and further research in this direction necessary.

Response : We have added two paragraphs in the Discussion section regarding disadvantages of current treatment approaches as well as possible reasons for superiority of the proposed drug combination.

 Point 4: Figures: Overall, figures are well presented. However, please make sure figure sizes (e.g. to have all significances (*) precisely over the bars) are similar in the different respective figures

Response : We corrected the significance symbol (*) in all representative figures.

Reviewer 3 Report

Authors present a manuscript demonstrating that the CAM assay could be employed as a preclinical in vivo model in CTCL for pharmacological testing. The paper is well-written, elegant and my opinion is that it may be accepted in its present form. 

Author Response

Response to Reviewer 3 Comments

Point 1: Authors present a manuscript demonstrating that the CAM assay could be employed as a preclinical in vivo model in CTCL for pharmacological testing. The paper is well-written, elegant and my opinion is that it may be accepted in its present form.

Response: We would like to thank the reviewer for the kind comments and the recommendation for the paper to be accepted in its present form.

Reviewer 4 Report

Dear Authors 

The paper is well done and the data found are very interesting

Only some minor- comments : 

The abstract is not set up like an abstract with a too colloquial style , make it more schematic , avoid phrases like  "in our previous study"

The discussion is too longer , summarize the first part of the text , giving more importance to the relevance of the results found 

Author Response

Point 1: The abstract is not set up like an abstract with a too colloquial style , make it more schematic , avoid phrases like  "in our previous study".

Response: We would like to thank the reviewer for the comment. We have revised the abstract and we added a GA abstract as well.

Point 2: The discussion is too longer, summarize the first part of the text, giving more importance to the relevance of the results found

Response : We have shortened this section as suggested.
